# Scale-Up of Pigment Production by the Marine-Derived Filamentous Fungus, *Talaromyces albobiverticillius* 30548, from Shake Flask to Stirred Bioreactor

Mekala Venkatachalam [1,2,*], Gary Mares [1,2], Laurent Dufossé [1,2] and Mireille Fouillaud [1]

1   Chemistry and Biotechnology of Natural Products (CHEMBIOPRO), Faculty of Sciences and Technologies, Université de la Réunion, F-97744 Saint-Denis, France
2   Ecole Supérieure d'Ingénieurs Réunion Océan Indien (ESIROI), 2 Rue Joseph Wetzell, F-97490 Sainte-Clotilde, France
*   Correspondence: mekalavenkat@gmail.com; Tel.: +33-1-847-807-9847

**Abstract:** *Talaromyces albobiverticillius* 30548, a marine-derived fungus, produces *Monascus*-like azaphilone red/orange pigments which have the potential for various industrial applications. The objective of this study was to scale up pigment production in a 2 L bioreactor with a working volume of 1.3 L media and to compare its biomass growth and pigment production against small volume (500 mL) shake flasks with 200 mL working volume. Additionally, fungal morphology, pigment intensity, fermentation length and duration of pigment production were also compared. Experiments were carried out at laboratory scale in 200 mL shake flasks without controlling pH and oxygen. In parallel, fermentation was performed in a 2 L bioreactor as an initial scale-up to investigate the influence of dissolved oxygen, agitation speed and controlled pH on pigment production and biomass growth of *T. albobiverticillius* 30548. The highest orange and red pigment production in bioreactor at 24 °C was noticed after 160 h of fermentation (70% $pO_2$) with 25.95 AU 470 nm for orange pigments and 22.79 AU 500 nm for red pigments, at pH set point 5.0. Meanwhile, the fermentation using 200 mL shake flasks effectively produced orange pigments with 22.39 AU 470 nm and red pigments with 14.84 AU 500 nm at 192 h under the same experimental conditions (24 °C, pH 5.0, 150 rpm). Regarding fungal morphology, growth of fungus in the bioreactor was in the form of pellets, whereas in the shake flasks it grew in the form of filaments. From the observed differences in shake flasks and closed bioreactor, it is known that the bioprocess was significantly influenced by dissolved oxygen saturation and agitation speed in scale-up. Thus, oxygen transfer appears to be the rate-limiting factor, which highly influences overall growth and production of pigments in *Talaromyces albobiverticillius* 30548 liquid culture.

**Keywords:** filamentous fungus; *Talaromyces albobiverticillius*; liquid submerged fermentation; bioreactor; pigments; morphology; pellets; filaments; agitation; dissolved oxygen; scale-up

## 1. Introduction

With an increase of consumer demands for natural origin in day-to-day products, the space for natural colors in the market continues to accelerate in its switch towards close-to-nature ingredients, which are phasing out artificial dyes. Having this focus, pigments produced by natural sources have become a point of interest among manufacturers and researchers for applications in the sectors of food, beverages, cosmetics, pharmaceuticals, dyeing clothes, paintings, inks, industrial coatings, etc. [1]. In the case of pigments, natural colors do not have the same color intensity as synthetics and some (not all) are less economical on a dosage basis; however, technological advances have reduced this performance gap. Therefore, production of natural colorants from various natural sources continues to rise due to its adequacy to meet consumer expectations, despite of its higher cost of production [2]. Among all available natural sources, pigment production from micro-organisms is

gaining interest over other sources. In particular, filamentous fungi have been recognized as the most promising potential source to produce not only an extraordinary range of pigments but also several bioactive compounds [3,4].

A wide range of fungi belonging to the genera *Aspergillus, Monascus, Penicillium, Periconia, Paecilomyces, Talaromyces* and *Trichoderma* are known to produce pigments of various hues and are widely studied [5,6]. Amongst all, a few potential candidates from the family of *Trichocomaceae* are key ones for industrial-scale production which includes *Aspergillus, Penicillium* and *Talaromyces* [3,7]. *Talaromyces albobiverticillius* (*Talaromyces* being the teleomorphic genus of *Penicillium* sp.) produces *Monascus*-like azaphilone pigments (MLAP) [8,9]. From the fungi *T. albobiverticillius* 30548 strain, a total production of 12 different compounds was described and among them, six different compounds have been identified by LCMS analysis [10]. Literature studies have also shown that some species of *Talaromyces* synthesize yellow- (monascorubrin and rubropunctatin) and red- (monascorubramine and rubropunctamine) colored pigments and among them, few species do not produce any mycotoxin along with these pigments [8,11–15]. This factor allows the biotechnological production of azaphilone pigments using *Talaromyces* spp. favorable for large-scale production. The pigments produced by *T. atroroseus, T. albobiverticillius, T. purpurogenus, T. aculeatus* and *T. funiculosus* are diffusing into the culture medium in submerged fermentation. However, submerged fermentation in small volumes is a first step to assess the critical parameters, resulting in better understanding of the process before scale-up to industrial scale with the goal of increasing production yields [16].

Scale-up from shake flasks to bioreactor is aimed at producing target compounds in large quantities and towards improving specific yields and product quality, if all the interactive factors are well controlled. As a means of getting higher yield, several operational parameters such as pH, dissolved oxygen, heat and mass transfer, mixing time, shear rate and agitation speed are the key influencers to make the fermentation successful [17,18]. However, in filamentous fungi it is challenging to obtain consistent and reproducible data due to the fungal behavior and morphology inside the bioreactor with the interaction of various factors [19–21]. While the basic parameters in fermentation remain the same for all end applications, ranges and their requirements should be modified in response to the requirements of fungal type and target metabolites production [16,22].

The primary objective of this study was to run the liquid fermentation in a 2 L bioreactor to understand the behavior of fungal growth and pigment production in a highly controlled, closed bioreactor. Previous experiments in shake flasks allowed us to obtain the optimal conditions to produce pigments with a fixed, controlled temperature, agitation rate and composition of the culture media [23]. In submerged fermentation using a 2 L bioreactor, parameters such as temperature, pH, dissolved oxygen and aeration rate were considered the initial key factors to monitor the biomass growth and pigment production by setting the above optimized factors as constant. This study was an initial approach to identify the interaction of variables, influence of aeration and agitation, pH control strategy and thereafter, additional efforts will be done to improve the cultivation conditions and to enhance the production in a further scale-up.

## 2. Materials and Methods

### 2.1. Organism and Maintenance

*Talaromyces albobiverticillius* 30548 has been isolated from a marine sediment source in Reunion Island, Indian Ocean area, and sampled from Trou d'Eau, on the external slope of the coral reef, at 17 m depth (21°06′22.11″ S, 55°14′15.78″ E) [24]. The fungus was grown on potato dextrose agar plates (PDA, Sigma-Aldrich, St. Louis, MO, USA) at 24 °C for a period of 7 days and it produced dark-red colored pigments between days 4 to 7. After 7 days of growth period, the agar plates with the solid culture were maintained at 4 °C for conservation as well as to sub-culture at regular intervals.

### 2.2. Inoculum Preparation

To make liquid seed culture, Potato Dextrose Broth (PDB, Sigma-Aldrich, St. Louis, MO, USA) was used as a growth medium. An amount of 100 mg of mycelial spores were taken from the fully grown agar plates and aseptically inoculated into 250 mL shake flasks (Erlenmeyer) containing 100 mL working volume of PDB medium. After the addition of spores, the culture was kept in a shaking incubator at 24 °C with an agitation of 200 rpm (Multitron Pro, Infors HT, Bottmingen, Switzerland) for a period of 72 h.

The cultivation for pigment production was composed of two stages: the first stage was preparing liquid seed culture that acts as pre-inoculum; the second stage was submerged fermentation in shake flasks (250 mL total volume) and small-scale bioreactor of 2 L total volume using seed culture.

### 2.3. Fermentation in Shake Flasks

Preliminary fermentation runs were done using 200 mL working volume of PDB medium in 500 mL shake flasks. The media was sterilized at 121 °C for 15 min and once it was cooled down to room temperature, the medium pH was adjusted to 5.0 under sterile conditions using 0.1 M HCl before inoculation. From 72 h seed culture, 1% ($v/v$) was added to flasks and those were kept in a shaking incubator at 24 °C and 200 rpm for 10 days. During the length of fermentation, the pigment production and biomass growth were monitored every 24 h at regular intervals and data was recorded. All the experiments were performed as triplicates and statistical analysis such as one-way ANOVA was carried out using SigmaPlot ver.10 (Systat Software Inc., San Jose, CA, USA).

### 2.4. Fermentation in Bioreactor

As a scale-up from shake flasks, fermentation was performed in a 2 L bioreactor (BIOSTAT® A PLUS, Sartorius Stedim Biotech, Goettingen, Germany) sealed with a stainless steel head-plate. This is a compact, autoclavable fermenter with integrated controls and measurement tools which makes it easy to use, as well as to transit from shake flasks for culturing microbes. The working bioreactor is equipped with temperature and pH control, dissolved oxygen probe and flat-bladed impeller for stirring (3-blade segment impeller, BB-8847398, Sartorius Stedim Biotech, Goettingen, Germany). A working volume of 1.3 L PDB medium was in situ sterilized at 121 °C for 15 min and upon cooling, it was inoculated with 5% $v/v$ of pre-inoculum (1.3 g/L wet mycelia). After the inoculation, there was no significant change in the working volume of the medium. The principal culture parameters such as temperature (24 °C) and initial pH (5.0) for this experiment were kept the same as those used for the shake flasks experiments. The pH was controlled automatically to be kept within a range of 4.9–5.1 using a pH control module (LH Fermentation Ltd., Stoke Poges, Reading, UK) equipped with a steam-sterilizable pH electrode (EF—12/120 K8-HM-UniVessel, Sartorius Stedim Biotech, Goettingen, Germany) by the automatic addition of 0.1 M NaOH.

Based on the results of preliminary tests in the bioreactor, agitation speed and air input were fixed from the start of this experiment and at a later stage, that had to be changed after examination of the culture evolution and morphology. During the entire fermentation process, the agitation speed was maintained by a flat-bladed impeller (200–1000 rpm) and sterile air was supplied at 1.3 L/min (100% $pO_2$, dissolved oxygen) with the use of air filters but varied (50, 70 and 90%) in the later runs depending on fungal growth demand.

### 2.5. Biomass Estimation

Fermentation was carried out for a total of 240 h and in between, an aliquot of sample (5 mL) was drawn aseptically using a sterile single-use syringe once every 24 h throughout the entire length of fermentation. The fermented broth was filtered using 48 μm Nitex filter cloth (Nitex 03-48/31, SEFAR AG, Heiden, Switzerland) to separate the fungal biomass and supernatant. The separated biomass was precisely weighed using an analytical weighing balance (Adventurer Pro AS214 d = 0.0001 g, Ohaus Europe GmbH, Greifensee, Switzerland)

and it was considered as the wet fungal biomass. To determine the dry biomass weight, the sample was dried in a hot air oven (SNB 100, Memmert, Schwabach, Germany) at 105 °C for 17 h and afterwards, it was kept in the desiccator for 30 min to get a precise weight [25].

### 2.6. Estimation of Pigments Absorbance

The daily collected supernatant considered as extracellular pigments was used to measure the pigment absorbance. The intracellular pigments were extracted from the biomass at the end of fermentation (day 8) using ethanol after washing the biomass twice with deionized water. Since this extraction is intensive for daily measurements and possess pigments with most orangish red hues, the focus was shifted mostly towards extracellular pigments, which is of interest with most red pigments. Hence, extracellular pigments were more favored as they have a mixture of pigments which have been previously studied and published [10]. The maximum absorption of the extracellular pigments was determined by scanning the colored extracts over the range of 200–700 nm (UV-vis area) using a microplate reader (Infinite® 200 PRO series, Tecan Life Sciences, Mannedorf, Switzerland) [21]. The maximum absorbance wavelength of extracellular pigments in the filtrate was estimated at 470 nm (orange pigments) and 500 nm (red pigments). The extracellular supernatant was diluted with distilled water prior to measuring the absorbance, which was done to keep the concentration within an acceptable range. The pigment concentration in the extracellular sample was expressed as absorbance units (AU) by taking into consideration the dilution factor and the volume of sample.

### 2.7. Color Characteristics

The remaining extracellular supernatant after pigment absorbance was used to measure CIELAB color coordinates using Spectrocolorimeter (CM-3500d Spectrocolorimeter, Konica Minolta, Tokyo, Japan). The CIE L*a*b* colorimetric system was interpreted as follows: the value L* indicates lightness and covers from 0 (black) to 100 (white). The positive to negative a* value indicates red to green colors, whereas positive and negative b* represents yellow or blue colors, respectively. Chroma, denoted by C, gives saturation or purity of color. Hue angles, h° denotes the degree of redness, yellowness, greenness, and blueness by locating at 0, 90, 180 and 270°, respectively. The values of L, a*, b*, C and h° were obtained automatically during analysis from the color data software called SpectraMagic™NX ((version 1.9, Minolta Co., Tokyo, Japan). The standard illuminant D65, now considered as a principal reference illuminant, was used in all colorimetric measurements to display average daylight. Initial calibration was done using control and blank for quantification and color analysis [26].

Chroma (C*) and hue angle (h°) were calculated from the following equations:

$$\text{Chroma } C* = \sqrt{(a*)^2 + (b*)^2} \tag{1}$$

$$\text{Hue angle } h° = \tan^{-1}\left(\frac{b*}{a*}\right) \tag{2}$$

## 3. Results

### 3.1. Importance and Regulation of pH for Pigment Production

Typically, the pH of the culture medium was found to have an effect on biomass growth and pigment production in most of the filamentous fungi [23,25,27,28]. To fix a certain pH set point in the bioreactor to cultivate *T. albobiverticllius* 30548, several experiments were previously carried out in shake flasks with minimal volume of culture media by varying the initial pH at different levels (3, 4, 5, 6, 7 and 8). As there was no inline control to automatically adjust pH for shake flask experiments, it was manually adjusted at regular intervals to control the pH to the desired level throughout the fermentation period. Results exhibited highest production of pigments on day 8 (192 h) for the cultivation performed with a pH set point of 5 (Tables 1 and 2). The pigment yield was 22.39 AU 470 nm and

14.84 AU 500 nm in terms of absorbance units, representing orange and red pigments, respectively.

**Table 1.** Maximum absorbance at 470 nm in shake flasks representing the region for orange pigments.

| pH | 3 | 4 | 5 | 6 | 7 | 8 |
|---|---|---|---|---|---|---|
| Day of maximum absorbance | 9 | 7 | 8 | 9 | 8 | 9 |
| Maximum absorbance (AU) ± Standard error | 6.74 ± 0.84 | 14.91 ± 1.03 | 22.39 ± 2.59 | 21.70 ± 1.66 | 10.14 ± 0.91 | 7.22 ± 0.33 |

**Table 2.** Maximum absorbance at 500 nm in shake flasks representing the region for red pigments.

| pH | 3 | 4 | 5 | 6 | 7 | 8 |
|---|---|---|---|---|---|---|
| Day of maximum absorbance | 9 | 7 | 8 | 8 | 8 | 9 |
| Maximum absorbance (AU) ± Standard error | 4.04 ± 0.36 | 13.29 ± 0.42 | 14.84 ± 1.93 | 13.42 ± 0.83 | 9.94 ± 0.35 | 7.41 ± 0.26 |

Secretion of pigments into the culture media was observed at nearly similar rates for the culture media set at pH 5.0 and 6.0, if considering the pigment absorbance at the wavelength of 470 nm (22.39 AU 470 nm ± 2.59; 21.70 AU 470 nm ± 1.66). For the wavelength at 500 nm, absorbance at pH 4.0 has a similar value (13.29 ± 0.42) compared to pH 5.0 and 6.0, whereas absorbance at 470 nm is significantly lower for pH 4 (14.91 ± 1.03). This showed that running the fermentation at a moderately acidic pH favored highest pigment production in *T. albobiverticillius* 30548, as noticed for many other strains such as *Penicillium purpurogenum* GH2 [15], *P. aculeatum* ATCC 10409 [27] and *Monascus* spp. [29–31]. However, absorbance of pigments with orange hues (maximum absorbance at 470 nm) seemed slightly lower compared with red pigment absorbance at 500 nm when the pH dropped to 4.0.

The suitability of the chosen pH values with respect to pigment production and its comparison of culture media with different pH were investigated using ANOVA SigmaPlot ver.10 (Systat Software Inc., San Jose, CA, USA). The maximum absorbance of pigments at 470 nm was obtained when conducting the experiments with the pH set at 5 and 6 (values are shown in Table 1). The confidence interval, both upper and lower limit, at 95% for 470 nm is ± 7.18 with pH 5 and ± 4.18 with pH 6. The confidence interval is a bit wider in the group with pH 5 due to high standard error from the triplicates, but the range was narrow with the samples from pH 6.

For red pigment production at 500 nm, the confidence interval at 95% was analyzed for three groups with pH 4, 5 and 6 since this pH range yielded higher pigment values (Table 2). Confidence interval at 95% was carried out to find the probability of repeating the true value range within each experiment set. The confidence interval (95%) at both upper and lower limit was calculated as ± 1.22, ± 5.63 and ± 2.42 for experiments with pH 4, 5 and 6, respectively. The interval at pH 5 for both wavelengths 470 nm and 500 nm was relatively high and this indicates that there should be considerable influence from many factors.

The results from shake flasks experiments were compared with values obtained from fermentation achieved in bioreactor. The pH set point was optimized and fixed at 5.0 to carry out the cultivation in the bioreactor of 2 L capacity.

*3.2. Changes in Fungal Morphology and Rheology*

To achieve successful fermentation with desired fungal morphology and targeted pigment yield, fungal morphology and concentration of inoculum volume are critical factors. To obtain desired morphology, controlled agitation speed and dissolved oxygen are considered the most crucial elements to be set, according to the need of fungi to grow and hence produce pigments [32]. Therefore, to analyze the impact of agitation speed and

dissolved oxygen rate on fungal morphology, two different fermentations in a 2 L bioreactor were performed using an agitation speed of 30 rpm with 100% $pO_2$ and 200 rpm with 50% $pO_2$. During the course of fermentation, the pH of the culture medium was automatically controlled with the addition of 0.1 M NaOH through tubing from the base line. Interestingly, two different types of fungal morphologies were observed in the submerged fermentation for the same fungal strain, but it was when the agitation speed was varied, which is depicted in Figure 1a, b.

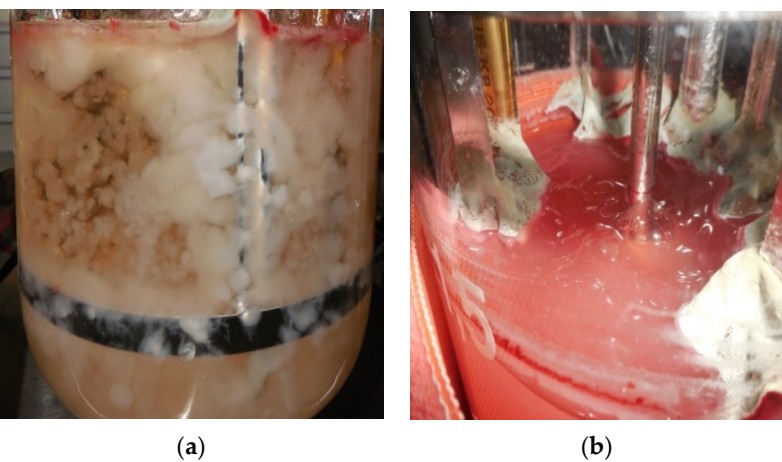

(**a**)          (**b**)

**Figure 1.** Stirred-tank 2 L bioreactor with pigments produced in the submerged fermentation of *T. albobiverticillius* 30548 in PDB, on day 7. (**a**) Agitation at 30 rpm with 100% $pO_2$, (**b**) agitation at 200 rpm with 50% $pO_2$.

In Figure 1a, it was observed that at an agitation rate of 30 rpm, the fungi grew in the form of pellets and looked fluffy, and visible pigments were observed in negligible amounts. As a result, culture broth viscosity was higher, thus inhibiting the dissolved oxygen to 0% after 16 h (Figure 2). On the other hand, when rotational speed of the impeller was set at 200 rpm, the growth of the fungus was observed as mycelia in the form of thin filaments (Figure 1b). Moderate agitation allowed the pellets to move freely in the liquid media and subsequently the pellets were broken down to form mycelia due to higher shear force. Also, the air was diffused above the surface of the liquid culture to maintain the concentration of dissolved oxygen in the liquid medium. On day 7, it was observed that the culture medium was homogenously red colored, exhibiting the production of pigments in the batch (Figure S1).

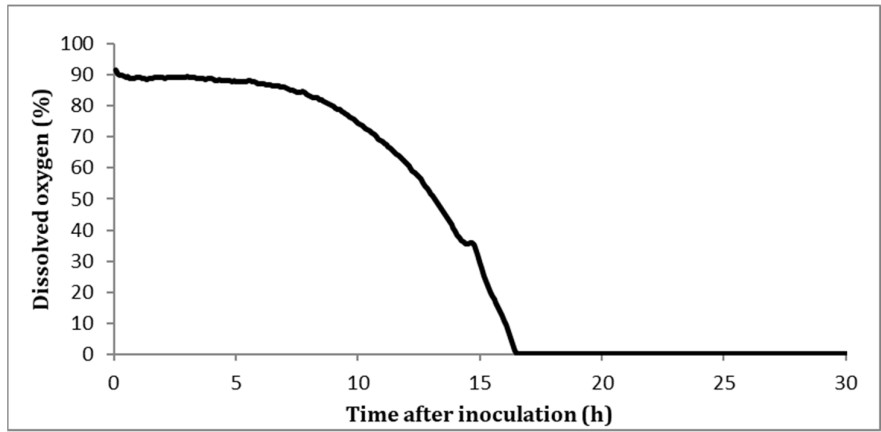

**Figure 2.** Decreased response of dissolved oxygen (0.2% air saturation) after fungal inoculation in 2 L bioreactor using 1.3 L working volume of PDB at 25 °C, with an aeration rate of 190 mL/min (90% initial air saturation) and stirring at 30 rpm.

In *Talaromyces albobiverticillius* 30548, two main types of pigments were of interest. Based on the absorption spectra, red pigments were produced in higher quantities and diffused into the extracellular culture medium. On the other hand, growing fungal mycelia produces orange pigments, which are considered intracellular pigments that are bound to the cells of the fungi and that can be extracted using polar solvents [33]. Even after changing parameters such as agitation speed and rate of oxygen supply to increase pigment production, pellet formation was a phenomenon observed in bioreactor but not in flask fermentation. To understand if the formation of pellets inhibits pigment production, 100 mL of medium containing pellets was drawn from the bioreactor (50% dissolved oxygen, 200 rpm) on day 5 and transferred into 250 mL Erlenmeyer flasks and Eppendorf tubes. These cultures were cultivated again for 5 days under shaking at 200 rpm without adjusting pH.

As can be observed in Figure 3a, the effect of oxygen transfer in shake flasks influenced red pigment production by increasing the absorption spectra in the visible region of 500 nm. On the contrary, in Eppendorf tubes, which are tightly closed to provide anaerobic conditions, there was no production of pigments (UV—visible scanning 200–600 nm) (Figure 3b). Hence, it was confirmed that fungal morphology in the form of pellets does not have a strong influence on pigment production. Contrarily, oxygen availability seemed to be the main influential parameter on pigment production noticed in this experiment.

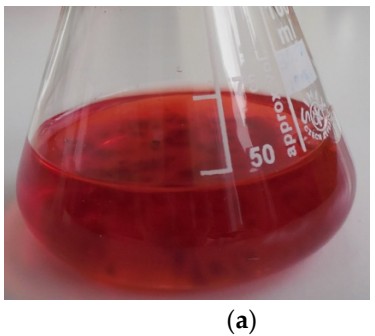 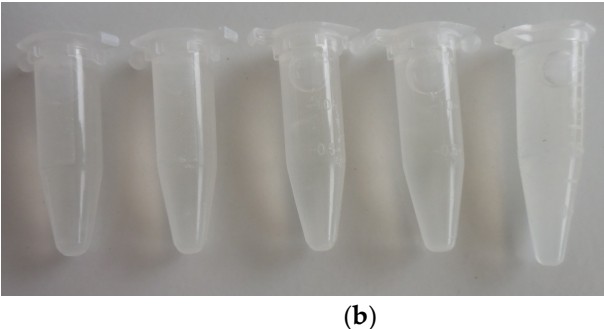

(**a**)                      (**b**)

**Figure 3.** (**a**) Pigment production from pellets of *T. albobiverticillius* 30548 using PDB media in shake flasks with silicone stopper on day 5; (**b**) growth of the same pellets in Eppendorf tubes on day 5 without any pigment production.

### 3.3. Combined Effect of Agitation Speed and Dissolved Oxygen on Biomass Growth

Agitation speed and oxygen supply are two crucial factors in pigment production as well for the growth of fungi [34]. To understand the interaction between them, three different fermentation runs were carried out. The level of dissolved oxygen ($pO_2$) was maintained at 50, 70 and 90% of saturation for runs 1, 2 and 3, respectively. Throughout fermentation at three different levels of $pO_2$, the agitation speed was found to change from time to time until the end of fermentation at 150 h.

In the first fermenter run, the dissolved oxygen was set at 100% with the agitation speed at 150 rpm. Initially, the parameters remained stable up to 10 h, but as soon as growth entered the exponential phase, dissolved oxygen dropped to 50% and simultaneously the agitation speed ramped up to 550 rpm. There was a steady maintenance of 50% dissolved oxygen throughout the end of fermentation at 160 h (Figure 4a,b). Regarding biomass growth, 5 g/L was obtained at 160 h. The biomass concentration in the two fermentation runs were similar and produced about 5 g/L when the agitation speed was kept at a moderate speed ($\leq$500 rpm), which is depicted in Figures 4b and 5b. At an agitation speed above 500 rpm and a higher air saturation at about 90%, the final biomass weight dropped from 5 to 4 g/L and is shown in Figure 6b.

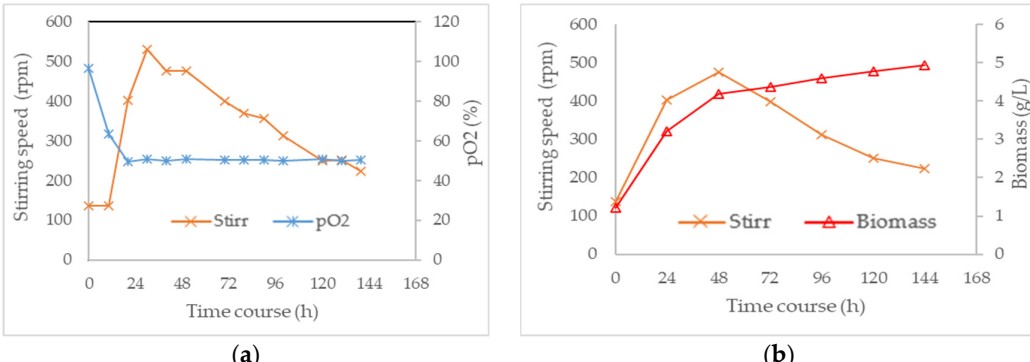

(a)                (b)

**Figure 4.** (**a**) Maintenance of 50% dissolved oxygen over time (0 to 168 h) in a submerged culture of *T. albobiverticillius* 30548 in a 2 L bioreactor showing the variation of stirring speed over time (0 to 160 h); (**b**) effect of agitation speed over time (0 to 160 h) on biomass production in a submerged culture of *T. albobiverticillius* 30548 in a 2 L bioreactor in combination with dissolved oxygen (pO$_2$) at 50%.

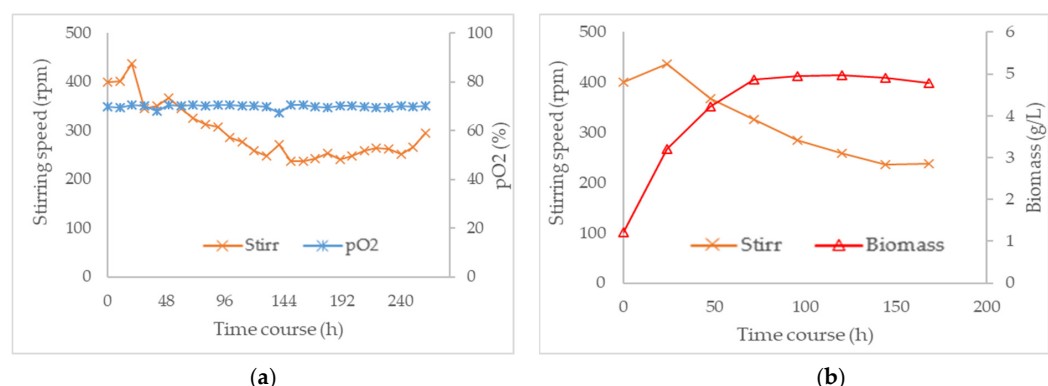

(a)                (b)

**Figure 5.** (**a**) Maintenance of dissolved oxygen at 70% over time (0 to 168 h) in a submerged culture of *T. albobiverticillius* 30548 in a 2 L bioreactor showing the variation of stirring speed over time (0 to 160 h); (**b**) effect of agitation speed over time (0 to 160 h) on biomass production in a submerged culture of *T. albobiverticillius* 30548 in a 2 L bioreactor in combination with dissolved oxygen at 70%.

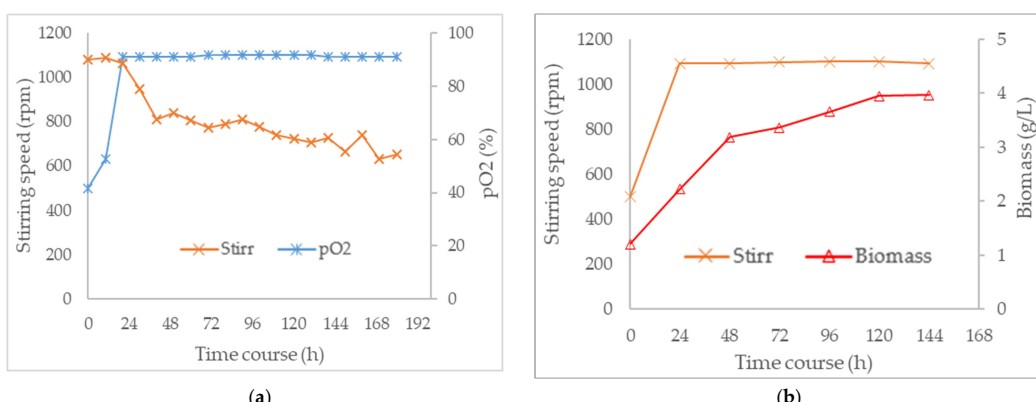

(a)                (b)

**Figure 6.** (**a**) Maintenance of dissolved oxygen at 90% over time (0 to 168 h) in a submerged culture of *T. albobiverticillius* 30548 in a 2 L bioreactor showing the variation of stirring speed over time (0 to 160 h); (**b**) effect of agitation speed over time (0 to 160 h) on biomass production in a submerged culture of *T. albobiverticillius* 30548 in a 2 L bioreactor in combination with dissolved oxygen at 90%.

*3.4. Effect of Agitation Speed and Dissolved Oxygen on Pigment Production*

In the initial fermentation at 10% saturated oxygen transfer, the fermentation broth became reddish orange in color after 4 days of cultivation and the maximum pigment

production was reached on day 8. When pO$_2$ was increased to 50%, the formation of pigments in the cultivation media began on day 4 (Figure 7). With an increase of pO$_2$ to 70% saturation, pigments were observed on day 2 after inoculation (Figure 8). Similar behavior was noticed with 90% pO$_2$ on day 2, but pigment production was more pronounced at 470 nm and 500 nm, indicating a mixture of pigment composition (Figure 9). Alternatively, the broth became red on day 3 in the culture fermented in shake flasks of 200 mL and the growth ended after 7 days due to the depletion of nutrients. On day 7, the observed pigment absorbance units were 20.38 AU 470 nm ± 0.43 and 18.23 AU 500 nm ± 0.31 in flasks. The production of pigments was exceeded in the 2 L bioreactor with 50% pO$_2$ (24.69 AU 470 nm) and 70% pO$_2$ (27.20 AU 470 nm) compared to shake flasks. Fermentation with 90% pO$_2$ produced absorbance values of 14.5 AU 430 nm and 18.01 AU 500 nm, which showed more pronounced production of yellow pigments with higher dissolved oxygen. The absorption range at different wavelengths for the three different experiments is shown in Table 3. The low growth rate in bioreactor at 10% pO$_2$ might be due to the low rate of dissolved oxygen, which thereby lowered pigment production.

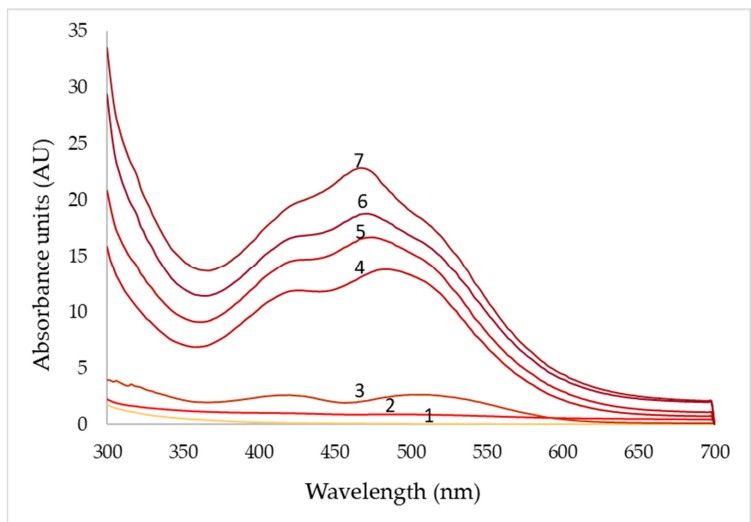

**Figure 7.** Absorbance scans of extracellular pigments (contained in the liquid broth) by spectropho­tometry produced from days 1–7 for the culture of *T. albobiverticillius* 30548 at 24 °C, pH 5, with pO$_2$ at 50%. Colored spectra indicates the production of pigments with respect to hues from day 1 to 7.

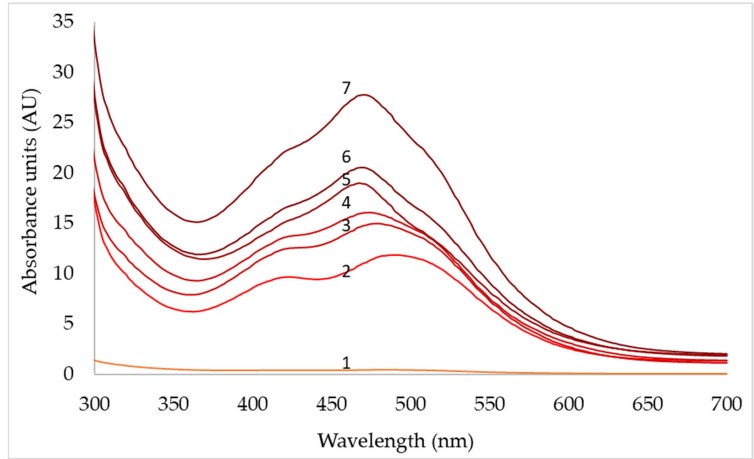

**Figure 8.** Absorbance scans of extracellular pigments (contained in the liquid broth) by spectropho­tometry produced from days 1–7 for the culture of *T. albobiverticillius* 30548 at 24 °C, pH 5, with pO$_2$ at 70%. Colored spectra indicates the production of pigments with respect to hues from day 1 to 7.

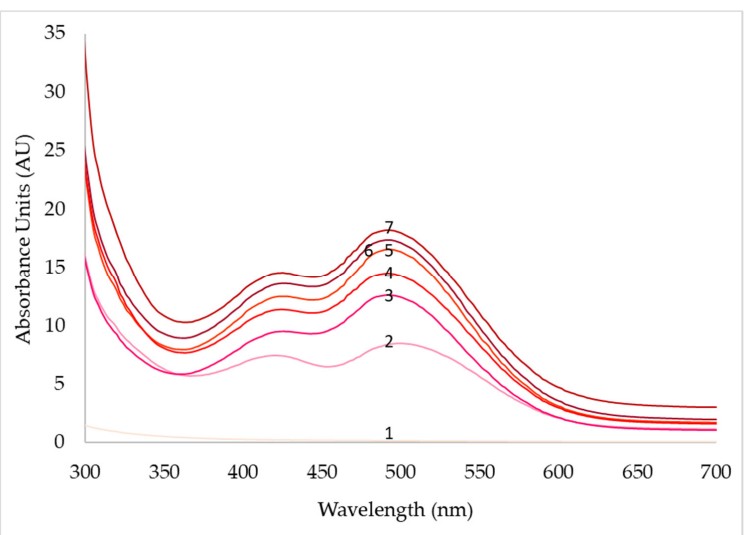

**Figure 9.** Absorbance scans of extracellular pigments (contained in the liquid broth) by spectrophotometry produced from days 1–7 for the culture of *T. albobiverticillius* 30548 at 24 °C, pH 5, with pO$_2$ at 90%. Colored spectra indicates the production of pigments with respect to hues from day 1 to 7.

**Table 3.** Maximum absorption of the extracellular pigments produced under different conditions using PDB culture media in a 2 L bioreactor measured at the wavelengths of 430, 470 and 500 nm indicating the regions for yellow, orange and red pigments, respectively.

| Days | pO$_2$ (50%) | | | pO$_2$ (70%) | | | pO$_2$ (90%) | | |
|---|---|---|---|---|---|---|---|---|---|
| | 430 nm | 470 nm | 500 nm | 430 nm | 470 nm | 500 nm | 430 nm | 470 nm | 500 nm |
| 1 | 0.1 | 0.1 | 0.1 | 0.4 | 0.2 | 0.2 | 0.3 | 0.2 | 0.2 |
| 2 | 0.9 | 1.0 | 1.2 | 9.5 | 11.3 | 11.6 | 7.4 | 7.1 | 8.4 |
| 3 | 2.3 | 2.2 | 2.8 | 12.5 | 15.1 | 14.4 | 9.2 | 11.1 | 12.3 |
| 4 | 12.2 | 13.5 | 134 | 14.6 | 16.0 | 14.6 | 11.4 | 13.2 | 14.8 |
| 5 | 14.7 | 16.8 | 15.4 | 16.4 | 19.2 | 15.0 | 12.7 | 15.4 | 16.6 |
| 6 | 16.9 | 19.6 | 17.4 | 18.2 | 20.7 | 16.6 | 13.6 | 15.6 | 17.4 |
| 7 | 20.2 | 24.6 | 19.2 | 22.8 | 27.2 | 23.5 | 14.5 | 16.4 | 18.0 |

*3.5. Colorimetric Characterization of Extracellular Pigments*

The culture filtrates obtained from cultivations of *T. albobiverticillius* 30548 at three different pO$_2$ levels were quantitatively characterized by colorimetric analysis by using CIEL\*a\*b\* coordinates. The culture filtrates contain the pigments excreted by the fungus during cultivation (extracellular pigments). In general, drastic changes in L\*, a\*, b\* coordinates were observed from day 1 to day 4 during the exponential phase for the three runs with varying levels of pO$_2$ (Table 4). In run 1 with pO$_2$ at 50%, once pigment production and fungal growth reached the stationary phase on day 4, there was no remarkable difference in color coordinates from day 5. Figure 10a shows that lightness value was in the range of 31 on day 7. The hue angles a\*, b\* of the pigments produced were in the positive range and ranged from 57–60 and 54–66, indicating more red and yellow hues, respectively.

In run 2 with pO$_2$ at 70%, L\* values dropped from 93.85 on day 1 to 51.88 on day 2, from indicating a decrease in brightness contributing to the darkness of color. This observation was contributed to increase in the production of red pigments and it could be attributed to the change of pigment composition based on the dissolved oxygen level and agitation speed maintained in the cultivation of this batch (values are specified in Table 5). Regardless of duration, the values of a\* and b\* remained in the range of 58–61 and 58–77, respectively, on days 2–7, showing that there was no remarkable change in the color hues (Figure 10b). This is in accordance with the absorbance value (27.20 AU) obtained at 470 nm in the absorbance scan done in the visible region representing the orange region (Figure 8).

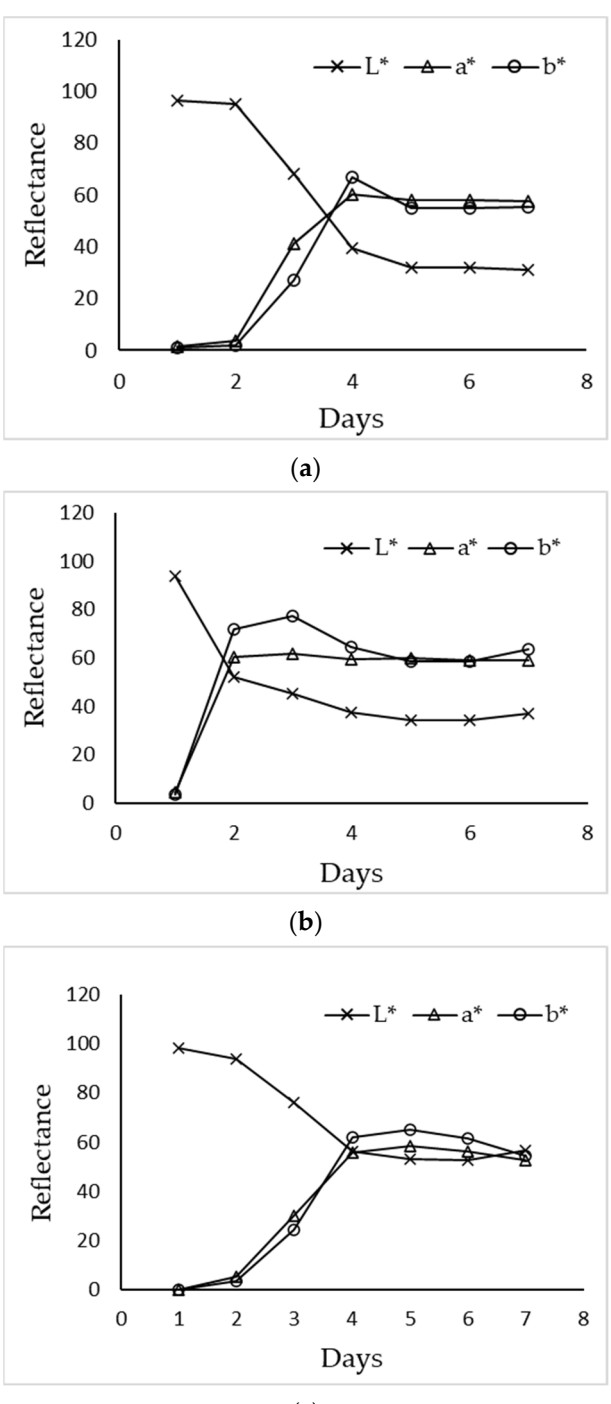

**Figure 10.** Color CIEL*a*b* values of pigments produced by *T. albobiverticillius* 30548 measured by colorimetric system (D65) with pO$_2$ set at (**a**) 50%, (**b**) 70% and (**c**) 90%.

In the fermentation with 90% pO$_2$, red and orange hues were observed, which is well identified from the coordinate values at day 7: L* (56.87), a* (52.86) and b* (54.61). From day 1 to day 4, there was a notable change in a*, b* coordinates similar to run 1, but not with respect to L* values (Table 6). In this run, lightness L* value remained around 56.87 on day 7, higher than the values obtained in run 1 (31.24) and run 2 (36.94). The difference in L* value is well understood from the spectral scan having absorbance at two regions, one at 430 nm and another at 511 nm, depicting the regions for yellow and red colors, respectively. As opposed to runs 1 and 2, it has absorbance values of 14.5 AU at 430 nm and 18.01 AU at 500 nm. Additionally, it was concluded that positive a*, b* values and the absorbance at

430, 470 and 500 nm are well correlated, which also confirmed the production of yellow, orange and red pigments accordingly, which is related to specific chromology (Figure 11).

**Table 4.** Colorimetric (D65) values L*a*b*c*h of extracellular extracts from 2 L bioreactor fermentation with $pO_2$ at 50%.

| | pO2 at 50% | | | | |
|---|---|---|---|---|---|
| **Days** | **L* (D65)** | **a* (D65)** | **b* (D65)** | **C* (D65)** | **h (D65)** |
| 1 | 96.36 | 1.56 | 0.87 | 3.13 | 152.43 |
| 2 | 94.91 | 3.75 | 1.85 | 4.66 | 151.57 |
| 3 | 68.26 | 41.39 | 27.11 | 49.49 | 33.33 |
| 4 | 39.44 | 60.13 | 66.77 | 89.87 | 48.02 |
| 5 | 31.76 | 58.21 | 54.93 | 80.04 | 43.35 |
| 6 | 32.07 | 57.81 | 55.01 | 79.81 | 43.59 |
| 7 | 31.24 | 57.54 | 55.30 | 79.81 | 43.86 |

**Table 5.** Colorimetric (D65) values L*a*b*c*h of extracellular extracts from 2 L bioreactor fermentation with $pO_2$ at 70%.

| | pO2 at 70% | | | | |
|---|---|---|---|---|---|
| **Days** | **L* (D65)** | **a* (D65)** | **b* (D65)** | **C* (D65)** | **h (D65)** |
| 1 | 93.85 | 4.36 | 3.71 | 5.75 | 36.98 |
| 2 | 51.88 | 60.31 | 71.75 | 93.88 | 49.65 |
| 3 | 45.21 | 61.84 | 77.24 | 99.08 | 51.15 |
| 4 | 37.53 | 59.60 | 64.48 | 87.92 | 47.13 |
| 5 | 34.25 | 60.08 | 58.73 | 84.01 | 44.35 |
| 6 | 34.03 | 58.96 | 58.40 | 83.00 | 44.70 |
| 7 | 36.94 | 58.82 | 63.42 | 86.52 | 47.15 |

**Table 6.** Colorimetric (D65) values L*a*b*c*h of extracellular extracts from 2 L bioreactor fermentation with $pO_2$ at 90%.

| | pO2 at 90% | | | | |
|---|---|---|---|---|---|
| **Days** | **L* (D65)** | **a* (D65)** | **b* (D65)** | **C* (D65)** | **h (D65)** |
| 1 | 98.05 | 0.24 | 0.3 | 0.38 | 52.08 |
| 2 | 93.71 | 5.48 | 3.74 | 6.63 | 34.3 |
| 3 | 76.16 | 30.1 | 24.6 | 38.88 | 39.26 |
| 4 | 56.08 | 55.97 | 62.1 | 83.6 | 47.98 |
| 5 | 53.24 | 58.43 | 65.08 | 87.46 | 48.08 |
| 6 | 52.59 | 56.16 | 61.51 | 83.29 | 47.61 |
| 7 | 56.87 | 52.86 | 54.61 | 76.01 | 45.93 |

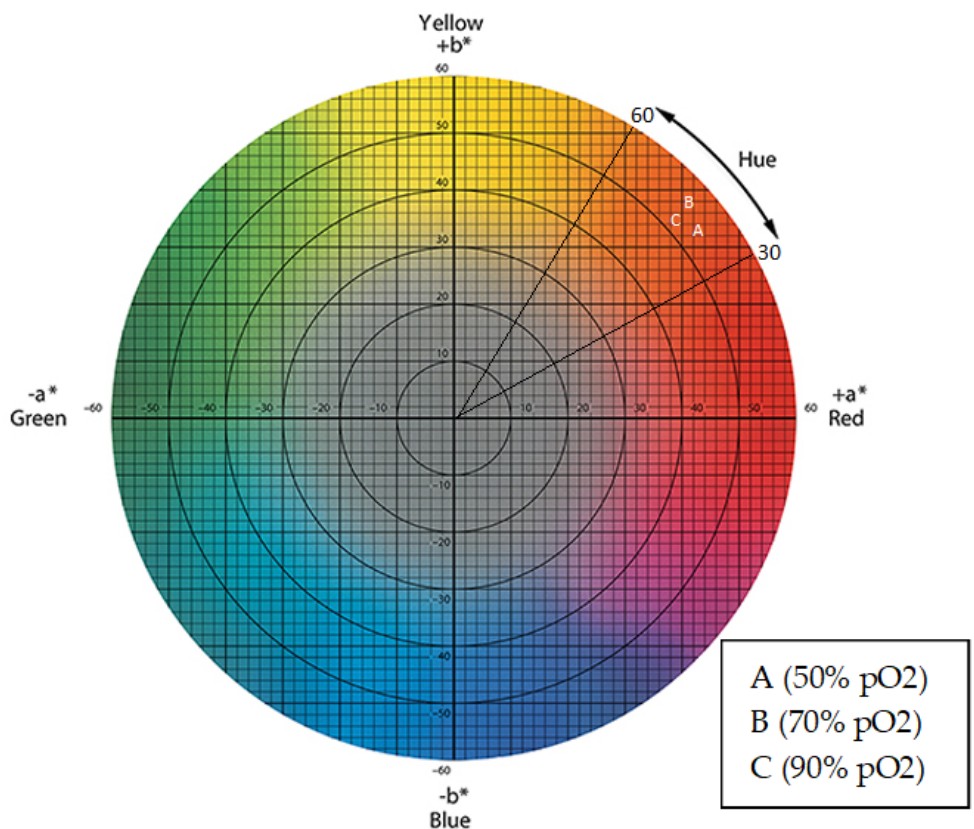

**Figure 11.** Colorimetric parameters in the color wheel describing the coordinate axis L*, a*, b*. The color hues are depicted as (**A**–**C**) for pO$_2$ at 50%, 70% and 90%, respectively.

An alternative way of specifying the color coordinates is using C*, h* values in place of a*, b*. Our data reported in Tables 1–3 indicated that all h* values were similar for the three different runs and in the range of 45 ± 2. As h* values are expressed in degrees, with 0° being a location on the +a* axis, 90° would be a point on the +b* axis, and from the obtained value (45 ± 2), it is clear that the pigment produced is described as red, indicating the color tone. The chroma coordinate (C*) explains the distance from the lightness axis and is around the value of 80 for all three runs towards the brighter shade, with 0 being at the center, denoting the grey shade. The concept of hue and chroma values agrees well with visual experience and for visual comparison, Figure 10c is appended here to have a correlation between color hues and values obtained using the CIEL*a*b* system.

## 4. Discussion

The scale-up process from laboratory to pilot level to industrial scale presents several inherent challenges. To achieve an effective bioconversion and produce targeted metabolites in a fermenter, some critical parameters such as homogenization of culture media, pH regulation, mass and heat transfer and dispersion of gas should be regulated throughout, until the end of fermentation depending on the organisms. All these parameters could be regulated with the help of agitation by choosing an impeller that meets the needs of the biological process specific to a particular organism [28]. Besides, agitation plays a crucial role in influencing fungal morphology and, in turn, morphology has an effect on pigment production and its yield, including extraction efficiency [35].

Differences in fungal morphology, such as growing as pellets or mycelia, in submerged fermentation have been noticed in many different filamentous fungi, i.e., *Aspergillus*, *Rhizopus* or *Penicillium* strains [36–39]. In most cases, the formation of pellets due to low agitation speed and high bubbling coming out from aeration may interfere with O$_2$ penetration, dissolution or substrate uptake and thus considerably influences the efficiency of target

product formation [40–42]. However, low agitation speeds may not generate enough turbulent flow to disperse the air bubbles too effectively into the media. In this study with *Taloromyces albobiverticillus* 30548, at a low agitation speed of 30 rpm, pigment production was dramatically depressed. In orbital shake culture flasks at 200 rpm using the same PDB as culture media, growth always took place in the form of mycelia, but pellets were observed in slow-stirred tank bioreactors under the same basic culture parameters (pH 5, temperature 24 °C). This could be partly due to lack of proper oxygen transfer in the culture media in the bioreactor and also to the type of shear forces produced by the stirrer. This leads to the growth of cells with different morphologies, either pellets or filaments, and eventually to a decrease in pigment yield in the case of pelleted-structured growth [43]. The same behavior was observed with *Monascus* fungi, in which cell morphology as pellets had an unfavorable effect on final pigment yield, which diminished pigment production [44]. As a side effect, the 2 L bioreactor has a larger surface area containing liquid media in contact with gas compared to shake flasks. Also, accessories of the fermenter (agitation impeller, foam control probe and pH and temperature control probes) offer static aerated surfaces. Therefore, the free filamentous fungi seemed to attach and grow on the probes and the wall, where $pO_2$ is maximum and feeding medium is regularly spilled.

While studying the impact of biomass and pigment production in *Talaromyces albobiverticillius* 30548, it was observed that at low agitation speed (30 rpm) and variable $pO_2$ rates (10–100% $pO_2$), pigment production was inferior and the fungus grew as pellets. Similar observations of fungal growth as pellets based on the influence of agitation rates have been reported on *Neurospora intermedia* and *Rhizopus oryzae* [37,40]. In addition, agitation speed played a major role in determining fungal morphology but when considering pigment production, the effect of dissolved oxygen concentration was greater than that of agitation speed. From observation of current research, at high-speed agitation (>1000 rpm for 8 days), the structure of fungal mycelia was not damaged.

In shake flasks, the pigment production starts after 48 h and is observed as light orange initially. Fermentation ends within 9 days with dark red pigment production. It has been suggested that red pigments are derived from orange precursors by a chemical reaction and are considered the first biosynthetic product [45,46]. A similar color shift was observed in this experiment with the absorbance scan of pigmented extracts of *T. albobiverticillius*, the first one near 430 nm and the second one at 511 nm; a few days later, the first one moved in the wavelength range from 430 nm to 470 nm.

During scale-up, there was a drastic decrease in pigment yield and biomass at 10% $pO_2$ compared to shake flask cultures. In those experimental conditions, the pigment yield obtained from the bioreactor was considerably low (4.89 AU 470 nm, 5.09 AU 500 nm, in 1.3 L PDB working volume, 10 $pO_2$ and 100 rpm agitation speed) compared to shake flasks (22.21 AU 470 nm, 18.61 AU 500 nm, 200 mL working volume and 200 rpm agitation speed). This might be partly due to the low oxygen diffusion which adversely affected growth (4.88 g/L in fermenter vs. 8.10 g/L in shake flasks) and therefore pigment production. Different pH levels influenced the physiology of fungi, conidial development and pigment synthesis. Reducing the pH inhibits the formation of conidia and increases pigment production, suggesting that the pH of the medium might affect the transport of certain media constituents such as glucose and nitrogen sources [47]. The pH can affect the activity of enzymes involved in the biosynthesis of pigments and research findings by several groups proposed that careful selection of pH influences the production of the predominant color component [48,49]. For *T. amestolkiae* cultivation in a chemically defined media along with MSG, deep yellow colorants were observed using neutral and basic pH, whereas deep red colors were seen with acidic pH [46]. This is in conformity with the red colorant production in *T. albobiverticillius* 30548.

## 5. Conclusions

The ability of *T. albobiverticillius* 30548 to produce a mixture of red, yellow and orange pigments in a controlled environment using a 2 L bioreactor was demonstrated in this work.

In parallel with fermentation, a few studies were conducted in shake flasks in order to understand the influence of different parameters contributing to pigment production and fungal growth and morphology. This study has demonstrated that pigment production is influenced by dissolved oxygen uptake of microorganisms. In addition, higher agitation speed at 800 rpm led to superior nutrient mixing inside the bioreactor, which in turn enhanced the availability of dissolved oxygen, which was maintained at 90% throughout the fermentation. Our experimental observation clearly explained that agitation speed significantly affects fungal morphology and growth, which in turn influence oxygen transfer. However, these results demonstrated that the scale-up was successful with controlled factors and that enhanced production of pigments was achieved in bioreactor compared to shake flasks.

**Supplementary Materials:** The following supporting information can be downloaded at: https://www.mdpi.com/article/10.3390/fermentation9010077/s1, Figure S1: Pigment production of *T. albobiverticillius* 30548 in a 2 L stirred tank bioreactor on day 7 using PDB growth media cultured at 24 °C of pH 5.0, under the controlled agitation of 200 rpm with 50% pO$_2$.

**Author Contributions:** Conceptualization, M.F. and L.D.; experiments execution, M.V. and G.M.; writing and original draft preparation, M.V.; data analysis, M.V. and G.M.; review, editing, visualization, M.F., M.V. and L.D. All authors have read and agreed to the published version of the manuscript.

**Funding:** The authors are grateful to the Regional council of Reunion Island for the financial support dedicated to research on microbial pigments.

**Institutional Review Board Statement:** Not applicable.

**Informed Consent Statement:** Not applicable.

**Data Availability Statement:** Not applicable.

**Acknowledgments:** The authors would like to thank Cathie Milhau from ESIROI for her technical support.

**Conflicts of Interest:** The authors declare no conflict of interest. The funders had no role in the design of the study; or in the collection, analyses or interpretation of data; or in the writing of the manuscript; or in the decision to publish the results.

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
