# Peer review of "Scale-Up of Pigment Production by the Marine-Derived Filamentous Fungus, Talaromyces albobiverticillius 30548, from Shake Flask to Stirred Bioreactor"

_fermentation, doi:10.3390/fermentation9010077_

Round 1

Reviewer 1 Report

Draft submitted by Mekala Venkatachalam et al. studied the conditions of T. albobiverticillius 30548 fermentation for pigment production and scaled up to a 2L bioreactor. The results showed that that pigment production is influenced by dissolved oxygen, and bacterial morphology and growth are influenced by agitation speed. Generally, the article is logical and idea make sense for actual industrial production. But before being published, several comments need to be considered and addressed. And there should be double check of the grammar for the draft.

1. It is suggested to explain pO2 and oxygen transfer rate separately.

2. The interactions between fungal growth and pigment production are not described in 3.3.

3. In 3.3, there is no explanation for figure 6.

4. It is suggested that the chroma and hue angles of the pigments produced during the fermentation at the three pO2 are indicated in Figure 13.

5. The data 800 rpm and its experimental results in the Conclusion were not mentioned in the above section.

6. To compare the pigment yields of shake flasks and 2L bioreactors, it is recommended to list the absorbance values of the pigments under various fermentation conditions in a table. The results are more visual.

7. Industrial production generally wants to produce a single pigment. It is proposed to investigate the control of fermentation conditions leading to the production of a single pigment, which would be of more practical production interest.

Reviewer 2 Report

 The authors provided some valuable results on fungal pigment production from engineering point of view. The design of experiments is appropriate to evaluate the effects of processing parameters on the fungal morphology and pigment productions, with some important conclusions on the effects of agitation on the fungal morphology and effects of oxygen level on the pigments production. Also the authors had demonstrated the feasibility and potential of scaling up the bioprocessing towards mass production. The manuscript was well written and organized well. I have several suggestions that the authors may consider addressing in the manuscript before publication in the journal of Fermentation.

Line 102, How do you measure the weight of the spores. 100 mg seems huge amount of spores. The spores should usually be quantified by the number of spores such as 1x107 per mL of medium.

Line 115, Does this mean the inoculation rate was 1% (v/v), such as 1 volume of seed culture in 99 volumes of fermentation medium?

Line 130, Again, does this mean the inoculation rate was 5% v/v, e.g. 65 mL seed culture in the total of 1.3 liter medium?

Line 131, Please explain why "no significant change", 5% v/v inoculation rate you should have noticible volume change in the reactor.

Line 134, Here should be a range, such as 4.9 to 5.1

Line 142, Please specify how the air was sterilized, filter?

Line 146, Please specify how the samples were withdrawn aseptically, did you use syringe?

Line 157, Please describe how you extract the intracellular pigments at the end of fermentation.

Line 158-160, Intensive in measurements shouldn't b a reason of not studying the intracellular pigments if they are more valuable. Please justify why the extracellular pigments are more focused than intracellular pigments, from perspective of scientific value and economic feasibility

Line 200, Statistical analysis at 95% confidence interval should be performed to see if there is significant difference (p < 0.05) for pH5 and pH6 at 470 nm and pH4, 5 and 6 at 500 nm.

Line 286-287, This sentence is not complete, please revise

Figure 6., Please explain why the agitation was not the same in Fig. 6 a and b. Also the pO2 concentration was higher than 100% which is not possible, please explain

Line 375-382, It would be better to combine Figure 10, 11, 12 into one figure with three sub-figures such as figure 10a, b, c.

Figure 13., This figure doesn't show information about the results of fermentation treatments. If it's just a method used for defining colorimetric parameters, it should be put into the method section.

Line 387-390, Is there a way to combine Table 4, 5, and 6 into one table so it's easier to compare the difference between each treatment?
